# Contribution of Physical Education to the Daily Physical Activity of Schoolchildren in Saudi Arabia

**DOI:** 10.3390/ijerph16132397

**Published:** 2019-07-05

**Authors:** Osama Aljuhani, Gavin Sandercock

**Affiliations:** 1Department of Physical Education, College of Sports Science and Physical Activity, King Saud University, Riyadh 4545, Saudi Arabia; 2College of Education and Arts, Northern Border University, Arar 1321, Saudi Arabia; 3School of Sport, Rehabilitation and Exercise Sciences, University of Essex, ESA.3.13 Colchester, UK

**Keywords:** accelerometer, cardiorespiratory fitness, moderate-to-vigorous physical activity

## Abstract

The positive contribution of physical education (PE) to daily physical activity (PA) has been documented in past studies. However, little is known about the contribution of PE to inactive and unfit schoolchildren’s PA. Therefore, the purpose of the present study was to examine the contribution of PE to the daily PA of schoolchildren, especially for inactive and unfit schoolchildren. Accelerometers were used to measure the PA of 111 boys (M_age_ = 13.6 ± 0.8 years) across 7 days. Moderate-to-vigorous PA (MVPA) was measured during PE classes and on school days with and without PE classes. To measure the time that schoolchildren spent on MVPA, the accelerometer count (i.e., ≥2296 counts/minute) was used. Schoolchildren spent 22% of PE class time in MVPA. Times spent in MVPA were 12.9, 14.7 and 14.8 minutes higher on PE days than on days without PE for all, inactive, and unfit schoolchildren, respectively. Results showed that 40% percent and 24% of the schoolchildren met the recommended levels of PA on PE days and days without PE, respectively. It is concluded that, since PE classes increase daily engagement in MVPA, especially among inactive and unfit schoolchildren, PE classes should be conducted on all school days.

## 1. Introduction

Insufficient physical activity (PA) is the fourth leading risk factor for premature mortality and is responsible for 6% of deaths globally [1]. The health benefits of PA are diverse; for example, physically active adolescents have better skeletal health, cardiorespiratory fitness, and favorable body composition [2,3,4]. Recent review indicated that frequent inclusion of higher levels of physical activity and less time allocated for sedentary behaviors are highly associated with better health-related quality of life in children and adolescents aged 3–18 years [5]. On the contrary, higher durations or frequencies of physical inactivity and sedentary behavior among children and adolescents (5–17 years) are associated with metabolic syndrome in adolescents [6], who are overweight/obese and have higher clustered cardiometabolic risk factors [7]. Longitudinal data showed that children and adolescents who were highly active between ages 9 to 18 would be active adults [8]. A review showed a positive effect of regular physical activity on executive functions, attention, and academic performance in schoolchildren [9]. To promote schoolchildren’s health, the World Health Organization (WHO) recommends that children and adolescents between the ages of 5 and 17 years should engage in at least 60 minutes of moderate-to-vigorous physical activity (MVPA) on a daily basis [1]. 

Children spend approximately 40% of their daily waking hours at their educational institutions on approximately 200 days of the year [10]; these figures suggest that schools are important settings in which health-related PA must be promoted [11,12]. Physical education (PE) is shown as an opportunity for schoolchildren to engage in PA [13,14]. However, low levels of PA during PE lessons have been reported. Hollis et al. [15] reviewed past studies that had measured PA using accelerometers (*n* = 5) and found that schoolchildren spent an average of 35% (25–44%) of their PE classes engaging in MVPA.

Physical education has been shown to provide children with the necessary skills and motivations to engage in lifelong activities [16,17]. Additionally, it has been suggested that PE could contribute to increasing schoolchildren’s daily PA and reducing the amount of sedentary time [18]. Previous studies suggest that schoolchildren engage in high levels of PA on days with PE classes [18,19,20,21], and PE contributes to 6–23% of their daily activity [18,20,21,22,23]. Calahorro-Canada et al. [24] found that schoolchildren who were relatively less fit were particularly more active on school days that included PE classes. Physical education classes were also found to stimulate schoolchildren to be more active on school days when PA opportunities are provided [23]. Such findings are contrary to the ActivityStat hypothesis, which suggests that children compensate for additional energy expenditure during PE classes by being less active at other times [25]. Furthermore, contrary to the ActivityStat hypothesis, children who are less active when they are not at school, also tend to be less active when they are at school (i.e., during recess, lunchtime and even PE classes) [26]. More information about the possible influence of PE on daily PA of less fit and inactive schoolchildren is important.

It is known that physical activity is modulated by local socio-geographical factors [27]. Previous studies using self-report and pedometer measures indicated that the majority of Saudi Arabian schoolchildren were inactive [28,29,30,31,32]. A recent study, using accelerometers, reported that Saudi Arabian girls did not meet the recommended daily MVPA [33]. Self-report may provide more details about the type, frequency, and duration of PA [34]. However, PA data collected by self-report among children and adolescents are questionable because of inaccurate assessment of the amount and intensity of PA [34]. The main output of pedometers is step count during walking and running [35]. The major advantages of using pedometers are that they are low cost and easy to use [34,35]. However, they do not accurately estimate PA intensity [35]. On the contrary, accelerometers have been shown as a reliable, objective, practical, non-invasive and accurate measure to quantify the volume and intensity of PA [34,35]. The daily activity levels (including PA that occurs during PE classes) of boys in Saudi Arabia have not been assessed using accelerometers. Therefore, our first aim was to assess schoolchildren’s PA levels during PE classes by objectively measuring the time that they spend engaging in MVPA. We also aimed to assess the contribution of PE to overall PA by comparing the daily PA levels of Saudi Arabian schoolchildren on days with and without scheduled PE classes; specifically, we intended to determine the contribution of PE to the attainment of the recommended duration of daily MVPA (i.e., 60 minutes), especially among physically inactive and unfit schoolchildren. 

## 2. Materials and Methods 

### 2.1. Participants 

Participants from four randomly selected middle schools that are situated in the northern region of Saudi Arabia were invited to participate in the present study. An initial sample of 123 schoolchildren (boys) aged 12–14 years (age 12, *n* = 38; age 13, *n* = 44; age 14, *n* = 41) who agreed to participate in the study constituted the study sample. Consent forms, which explicated the requirements of the study, were provided to schoolchildren and their parents or guardians. All measurements were conducted in March and April 2017. Ethical approval for this study was obtained from the Northern Border University Ethical Review Committee (the ethics code is (7/38/H)).

### 2.2. Instrumentation and Procedure

The participants’ height and weight were measured to the nearest ± 0.1 cm using a Seca 213 Portable Stadiometer (Hamburg, Germany) and to the nearest ± 0.1 kg using a Digital Scale (model 770; Seca, Hamburg, Germany) respectively. All measurements were recorded by the principal investigator; the schoolchildren were barefoot and wore light clothing. The body mass index (BMI) was calculated using the following formula (kg/m²) and converted to z-scores using the LMS method [36]. Subsequently, participants were classified into four categories in accordance with the International Obesity Task Force (IOTF) criteria: underweight, normal weight, overweight and obese [37].

In Saudi Arabian middle schools, PE classes are typically provided once a week for a duration of 45 minutes and are taught by specialized and certified teachers. In the present study, teachers were asked to conduct a standard PE class in accordance with the national PE curriculum. Class timetables were collected after the measurements had been recorded. Participants’ PA levels were measured using an ActiGraph 3-axis accelerometer (wGT3X-BT, ActiGraph LLC, Pensacola, FL). They were monitored across seven days, and it is noteworthy that only the data collected during the weekdays were analyzed in this study. The resultant data were initialized, downloaded, and analyzed using ActiLife v6013.3 (ActiGraph LLC, Pensacola, FL). The data were collected at the beginning of the first day of measurement and downloaded on the seventh day. During the first visit on day 1, accelerometers were attached to participants’ right hips using an elastic belt. Schoolchildren were instructed to wear the accelerometers at all times (i.e., from morning until bedtime, except when they bathed or swam). Accordingly, data were recorded between 7:00 a.m. and 10:59 p.m. across seven days. Accelerometers were collected during the final visit on day 7; subsequently, as recommended by previous studies that examined children and adolescents, the raw data were downloaded with 1-second epochs [38,39]. The downloaded data were reintegrated into 60-second intervals in order to facilitate comparison with the findings of previous studies. As recommended by Trost et al. [39], established accelerometer cut-off points proposed by Evenson et al. [40] were used to define the PA levels (count per minute): sedentary (0–100), light (101–2295), moderate (2296–4011), and vigorous (≥4012).

Participants’ cardiorespiratory fitness (CRF) was measured using a modified version of the 20 meters shuttle run test (20 m SRT) in the form of FITNESSGRAM PACER [41]. Schoolchildren with medical conditions (*n* = 3) were excluded from the CRF test. This test was conducted during PE classes at the respective school’s indoor sports hall, and it was supervised by the principal investigator and the school’s PE staff. The 20 m SRT was conducted one week before PA was assessed using accelerometers. A maximum of five participants were assessed at a given point of time. In this test, participants were required to continuously run back and forth between two lines that were separated by a distance of 20 m for as long as possible; they were required to follow the instructions of the researchers as well as the recorded instructions on the PACER CD. The initial testing speed was 8 km/h and it was systematically increased by 0.5 km/h for each subsequent minute. Researchers recorded the final shuttle count when the participants failed to maintain the required speed across two consecutive rounds or when they had reached their volitional exhaustion points. The final shuttle count was expressed in terms of running speed and subsequently transformed into z-scores [42]. Maximal oxygen uptake ($\overset{\cdot }{V}$O_2max_) was estimated based on the final running speed according to a published equation [43]. The lower FITNESSGRAM PACER Healthy Fitness Zone was used to identify physically fit participants who met at least the minimum amount of sex- and age-specific cut-offs shuttle run; the remaining participants were considered to be physically unfit [41].

All data were managed and processed using Microsoft Access and Microsoft Excel 2013. In the present study, only data that were collected during the weekdays were included. The data were included in the analysis if participants wore the accelerometers for a minimum of three weekdays and for at least 600 minutes per weekday [44]. Non wear-time was classified as 20 minutes of non-compliance (i.e., failure to wear the accelerometer) [45]. Out of a total of 123 participants, 111 (90.3%) wore the accelerometer for a minimum of three days (at least one of which included PE classes). Twelve (9.7%) participants who did not meet the minimum inclusion criteria or with medical conditions were excluded from the study. Average daily MVPA (minute/day) was calculated by adding the total time that was spent on moderate physical activity (MPA) and vigorous physical activity (VPA) and dividing the resultant sum by the number of valid days. Daily MVPA was independently computed for days that had and did not have PE classes (i.e., average MVPA on days with and without PE classes). A cut-off of 60 minutes of daily MVPA, which is the recommended duration, was used to differentiate between active and inactive children. Schoolchildren who achieved the recommended MVPA on less than 50% of the valid days were categorized as inactive children, whereas those who achieved the recommended MVPA on 50% or more of the valid days were classified as active children [46,47].

### 2.3. Data Analysis

Descriptive characteristics (M ± SD) were computed, and an independent *t*-test was used to examine group differences in anthropometric data between active and inactive, and fit and unfit participants. In this study, three multilevel analyses were conducted using linear mixed models: days of measurement (Level 1), schoolchildren (Level 2), and school (Level 3). The first analysis was independently conducted for each of the two groups that differed in the level of PA (i.e., active and inactive). The dependent variable was time spent on MVPA (in minutes). Days with and without PE classes were coded as follows and included as fixed factors: 0 = days without PE classes (Day no-PE), 1 = days with PE classes (PEday+), 2 = PE days excluding PE time (PEday−). The school emerged as the smallest cluster and was therefore included as a random factor. The results of the analysis yielded parametric estimates (*β*, 95% confidence intervals (95% CI)) that were used to examine differences in MVPA between days with and without PE classes. The second analysis was independently conducted for the two groups that differed in their level of physical fitness (i.e., fit and unfit) in a manner that was similar to that of the first analysis. The same models were used to examine the contribution of the duration of time that participants spent on MVPA during PE class (PE-MVPA) to daily activity; the ratio of PE-MVPA to daily MVPA were the dependent variables. All analyses were adjusted for age, BMI, and CRF z-scores. The Cochran’s Q test was conducted to compare the percentages of participants who achieved the recommended duration of daily MVPA (i.e., 60 minutes) on days with and without PE classes; subsequently, post-hoc analysis, namely, McNemar’s test, was performed to examine the significant results of the Cochran’s Q test. All analyses were conducted using version 24 of IBM SPSS (IBM, Armonk, NY), and the statistical significance of the results was tested at an alpha level of 0.05.

## 3. Results

The descriptive statistics for the study variables, which were computed using data that were obtained from the 111 schoolchildren (75% inactive and 73% unfit) who met our inclusion criteria, are shown in Table 1. Active and inactive schoolchildren did not significantly differ on anthropometric measures or in PE-MVPA. However, when compared to inactive schoolchildren, active schoolchildren spent more time on daily MVPA (M_1_ − M_2_ = 33.4, 95% CI = 26.4, 40.3), obtained a higher CRF z-score (M_1_ − M_2_ = 0.4, 95% CI = 0.2, 0.6), and were faster during the 20 m shuttle run (M_1_ − M_2_ = 0.5, 95% CI = 0.2, 0.7). Similarly, as can be seen in Table 1, fit schoolchildren were younger (M_1_ − M_2_ = −0.6, 95% CI = −1.0, −0.3), had a lower body weight (M_1_ − M_2_ = −5.8, 95% CI = −10.8, −0.9) and BMI (M_1_ − M_2_ = −2.0, 98% CI = −3.1, −0.9), and a higher mean CRF z-score (M_1_ − M_2_ = 0.7, 98% CI = 0.6, 0.9) than unfit schoolchildren. Fit and unfit groups did not differ in PE-MVPA and daily MVPA.

### 3.1. Physical Activity on Days with and without PE Classes

The estimated marginal means for time spent on MVPA are shown in Figure 1 and Figure 2; across the three multilevel analyses, the effects of school clusters (random effect), and age and BMI (fixed effects) were adjusted. In general, schoolchildren spent more time on MVPA (M_1_ − M_2_ = 12.9, 95% CI = 7.1, 18.6) on PEday+ than on Day no-PE. Inactive schoolchildren spent more time on MVPA (M_1_ − M_2_ = 14.7, 95% CI = 8.9, 20.6) on PEday+ than on Day no-PE (Figure 1); however, the time that active schoolchildren (M_1_ − M_2_ = 7, 95% CI = −4.2, 18.3) spent on Day no-PE and PEday+ was not significant. After excluding the duration of the PE class from PE days (PEday−), no significant difference emerged in the time that schoolchildren spent on MVPA on PEday- and Day no-PE; this was true for the total sample (M_1_ − M_2_ = 2.9, 95% CI = −2.9, 8.9), active schoolchildren (M_1_ − M_2_ = −3.7, 95% CI = −15, 7.5), and inactive schoolchildren (M_1_ − M_2_ = 5.1, 95% CI = −0.7, 10.9). 

Unfit schoolchildren spent more time engaged in MVPA (M_1_ − M_2_ = 14.8, 95% CI = 8.3, 21.3) on PEday+ than on Day no-PE (Figure 2). A comparative analysis of PEday+ and Day no-PE showed no significant differences in the time that fit schoolchildren spent in MVPA (M_1_ − M_2_ = 7.6, 95% CI = 3.5, 18.6). Excluding MVPA from PE days, there was no significant difference in the time that fit (M_1_ − M_2_ = −2.7, 95% CI = 13.8, 8.3) and unfit schoolchildren (M_1_ − M_2_ = 4.9, 95% CI = 1.5, 11.4)) spent on MVPA. 

### 3.2. The Contribution of PE to Overall MVPA

The results of age, BMI, and CRF z-score-adjusted multilevel model showed that PE contributed to 21% of the variance in overall daily MVPA; the relative contribution was higher for inactive schoolchildren (23%) than for active schoolchildren (16%; Figure 1). The model also showed that PE contributed to the overall daily MVPA of fit (20%) and unfit schoolchildren (22%; Figure 2). Figure 3 shows the number of schoolchildren who were active and inactive on PE days and days without PE classes. The results of the Cochran’s Q test revealed significant differences in the percentage of schoolchildren who achieved the recommended duration of daily MVPA (i.e., 60 minutes) across Day no-PE, PEday+, and PEday−. Further, the results of a post-hoc McNemar’s test showed that 40% of the schoolchildren met the recommended duration of daily MVPA on PEday+; the percentages were lower on Day no-PE (24%, *p* = 0.005) and on PEday− (28%, *p* < 0.001). The results also showed that 25 schoolchildren (i.e., 22.5% of the study sample) who were active on PEday+ were inactive on Day no-PE. 

## 4. Discussion 

The present study showed that Saudi Arabian schoolchildren engaged in MVPA for 22% of the duration of a PE class. Further, PE significantly contributed to overall MVPA. Twenty-four percent of the schoolchildren met the recommended PA levels on days without PE classes, but 40% of the schoolchildren did so on days with PE classes. 

We found that Saudi Arabian schoolchildren spent one-fifth (22%) of the duration of a PE class on MVPA activities; this is consistent with reports that time spent on MVPA accounts for 12.9–68.2% of the duration of a PE class [15]. This mean estimate is below the lower limit of the confidence interval that has been reported in a meta-analytic review of methodologically comparable studies (*n* = 5) that have measured MVPA using accelerometers; specifically, the findings suggested that schoolchildren spent 34.7% of the duration of their PE classes on MVPA (95% CI = 25.1, 44.4) [15]. The proportion of PE class durations that Saudi Arabian schoolchildren spent on MVPA is comparable to those that have been observed in studies that have been conducted with Estonian (28.6%) [48] and Swedish (25%) schoolchildren [49]. We used a 1-second epoch to collect data (past studies have used 10- and 15-second epochs) because children engage in short bursts of MVPA, which are equivalent to 3–6-second epochs [50]. Given that shorter epochs result in higher MVPA estimates, our findings suggest that PE classes are less physically demanding in Saudi Arabia than in Estonia and Sweden.

In the United States of America, it has been recommended that 50% of PE class duration should be spent on MVPA [51]. Similarly, in the United Kingdom, the Association for Physical Education (AfPE) has recommended that schoolchildren should be physically active for at least 50% of the available learning time [52]. While the current findings are not surprising when considering those of previous studies, it is worrying that none of our participants met the recommended duration of MVPA during PE classes (i.e., 50% of the PE class duration); this finding is disappointing, when it is compared to those that have been reported in previous studies (e.g., 8%, 26%, and 42%) [21,48,49]. Low levels of MVPA that were observed in this study may have resulted from a lack of motivation [49]. Indeed, it has been suggested that activities must be fun and enjoyable if they are to effectively increase the MVPA of schoolchildren [53]. Additionally, it is likely that the teachers who were involved in this study experienced difficulties in helping schoolchildren engage in sufficient levels of PA. One such difficulty that PE teachers face pertains to the variety of PE-lesson objectives. Saudi Arabian middle schools aim to promote the health-related knowledge and spiritual, moral, social, and leadership skills of their schoolchildren; additionally, schoolchildren are expected to learn about the health-related components of physical fitness during PE classes. Moreover, these schoolchildren are also expected to be proficient in sports skills and sport-specific rules. Such goals are not incompatible with the recommendation that schoolchildren must spend 50% of their PE class duration on MVPA; however, the complexity and diversity of the PE curriculum make the attainment of such a goal difficult. One solution is to create professional development programs that are targeted toward PE teachers [54].

Our findings are similar to those of previous studies [20,21,48,55,56], which have found that schoolchildren spend more time on MVPA on days that include PE classes than on days that do not include PE classes. Indeed, in this study, 40% of schoolchildren met the recommended level of PA on days that included PE, but only 24% of schoolchildren met the recommended level of PA on days that did not include PE. Contrary to the ActivityStat hypothesis [25], we found no evidence to support the contention that schoolchildren decrease their activity levels to compensate for their additional energy expenditure during PE classes. 

An additional 11 minutes daily MVPA could reduce cardiovascular risk and increase aerobic fitness [57] and improve schoolchildren’s academic performance and attention [9,58]. Therefore, the additional 12.9 minutes of MVPA that was observed in this study appears to be clinically important. However, when time spent on MVPA during PE classes was excluded from the analysis, there was no difference in daily MVPA on PE days and days without PE classes.

The aforementioned finding must be interpreted with caution (i.e., additional time spent on MVPA) because the respective measurements were recorded on only one day. The benefits of PA that have been reported by Kriemler et al. [57] are based on increases in mean levels of PA across a period of more than five days. 

The additional time that is spent on PA is most beneficial to individuals with the lowest levels of initial PA. In this study, half of the schoolchildren who met the recommended levels of PA on PE days were inactive on days without PE. Thus, PE may stimulate children, especially inactive children, to be more active on PE days [55]. Previous studies have revealed that PE classes account for 15–18% of the daily MVPA of the least active schoolchildren [23,55].

PE classes provide important benefits (e.g., increased daily MVPA; 11.4%) to children with low aerobic fitness [24]. On the contrary, we found that fit and unfit schoolchildren were similar in the time that they spent on MVPA on PE days; further, PE contributed to 17% of daily MVPA. The contribution of PE to the overall MVPA of active (14%) and fit (17%) schoolchildren is in the expected direction; further, these results are in agreement with recent findings that PE contributes to 10–19% of daily MVPA [20,21,56]. Direct comparisons of the present and past findings are problematic because there are methodological variations in the cut-off values and accelerometer wear time that have been measured both during and after PE classes. For example, previous studies used ≥ 2000 and ≥ 2333 count per minute to defined time spent in MVPA, whereas in the current study ≥ 2966 count per minute was used. The contribution of PE classes to the overall MVPA of inactive (20%) and unfit (17%) schoolchildren and the absence of evidence that supports the ActivityStat hypothesis suggest that increasing the frequency of PE classes is a promising and simple interventional strategy that can increase the PA of school children. It also appears that PE may play an important role in improving children’s health by reducing the time that they spend on sedentary activities [48]. Lunch break time and classroom-based activities seem like promising opportunities for schoolchildren to accumulate more levels of PA during school time.

### Limitations and Suggestions for Future Research

The present study has various limitations. The first limitation pertains to the fact that we did not examine the contents of the PE lesson. Indeed, it has been documented that differences in the content of PE lessons differentially influence the time that schoolchildren spend on MVPA. It has been reported that team games show higher levels of PA than individual games [59]. Accelerometers are considered to be reliable and valid instruments that can be used to assess the PA levels of children and adolescents. However, they do not accurately assess some activities (e.g., those that involve upper-body movements) and may consequently lead to underestimates of PA levels. To overcome these limitations, future research studies must examine the contents of PE lessons. Another limitation of the present study is that all the participants were boys, and they were sampled from a small town in the northern regions of Saudi Arabia; therefore, the findings cannot be generalized to other demographic groups. Furthermore, the actual time that schoolchildren spent on PA during PE classes was not reported; therefore, the results may be biased because we did not adjust our model to account for the actual time that schoolchildren spent on PA during PE classes. When compared to previous research studies, our study is novel in that we controlled for the effects of participants’ fitness levels. This allowed us to conclude that the differences in PA that were observed on PE days and days without PE are not attributable to differences in fitness levels. However, these findings must be interpreted with caution because of the limitations of the measurement methods and the cross-sectional research design that the present study entailed. 

## 5. Conclusions

The findings of the present study revealed that Saudi Arabian middle school children do not meet the recommended levels of PA. However, PE contributed significantly to overall MVPA, and schoolchildren spent more time on MVPA on PE days than on days that did not include PE. Furthermore, the number of schoolchildren who spent the recommended duration (i.e., 60 minutes) of time on daily MVPA was higher on PE days than on days without PE. In the present study, PE increased children’s PA, especially those who are inactive and unfit. In contrast, the percentage of time that schoolchildren spent on PA during PE classes was below the recommended duration (i.e., 50% of the PE class duration). In conclusion, given that PE can increase schoolchildren’s overall MVPA, PE classes must be offered on all school days.

### Implications for Physical Education in Schools 

Although PE is taught by specialized teachers, the percentage of PE class duration that they devote to helping schoolchildren engage in MVPA is relatively low. Therefore, PE teachers should be encouraged to increase the time that schoolchildren spend on MVPA. Schools should ensure that PE teachers receive professional training so that quality PE is delivered. The results of the present study showed that PE significantly contributes to overall MVPA and that the overall MVPA of inactive and unfit schoolchildren increased when they engaged in PE classes. Furthermore, half of the schoolchildren who met the recommended levels of PA on PE-days were inactive on days without PE. However, international recommendations emphasize that children and adolescents should be active for at least 60 minutes per day. Thus, the additional 13 minutes of MVPA that was recorded on PE day (i.e., when compared to time spent on MVPA on days without PE) indicates that PE can be effective in improving schoolchildren’ PA levels if PE classes are offered on all school days. 

## Figures and Tables

**Figure 1 ijerph-16-02397-f001:**
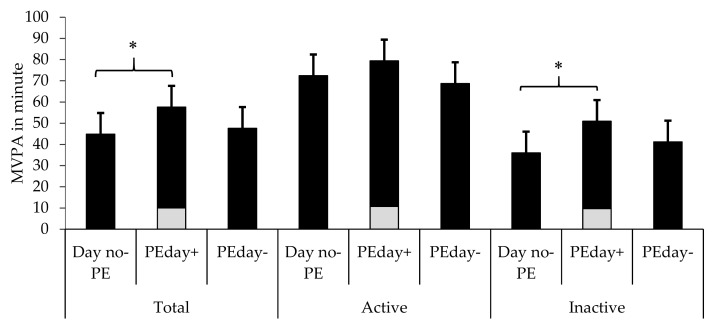
Adjusted time spent on MVPA on Day no-PE, PEday+, and PEday− by the total sample, and by active and inactive schoolchildren. Gray squares refer to MVPA in PE. *Significant differences between groups.

**Figure 2 ijerph-16-02397-f002:**
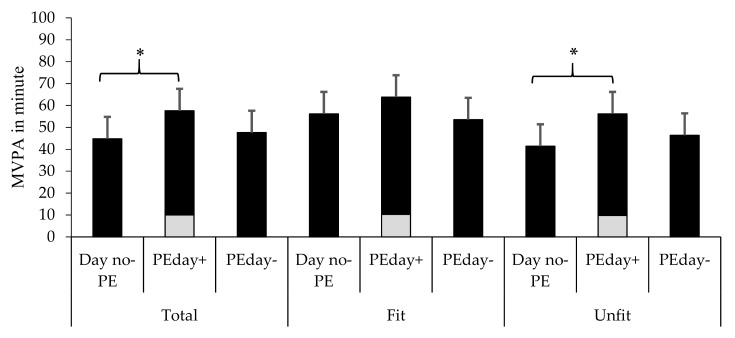
Adjusted time spent on MVPA on Day no-PE, PEday+, and PEday−, by the total sample, and fit and unfit schoolchildren. Gray squares refer to MVPA in PE. *Significant differences between groups.

**Figure 3 ijerph-16-02397-f003:**
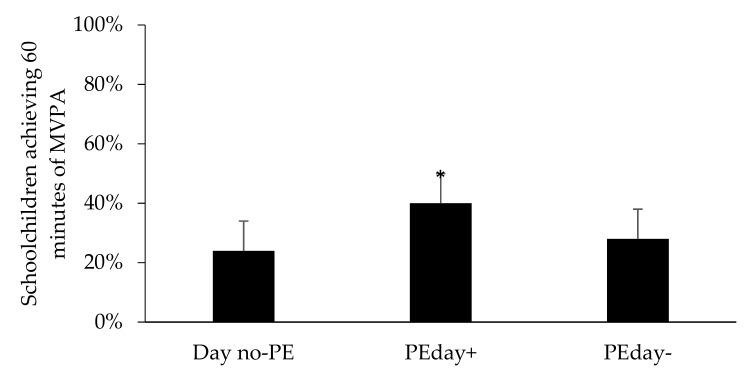
The percentages of schoolchildren who met the recommended duration of daily MVPA (i.e., 60 minutes) on Day no-PE, PEday+, and PEday−. * Significant differences to Day no-PE (*p* = 0.005) and PEday- (*p* < 0.001).

**Table 1 ijerph-16-02397-t001:** Descriptive statistics for the study variables across groups that differed in their levels of physical activity and fitness.

Variable	Mean (SD)					
	Activity Status			Fitness Status		
	Total Sample	Active	Inactive		Fit	Unfit
	*n* = 111	*n* = 27	*n* = 84		*n* = 29	*n* = 82
Age (years)	13.6 (0.8)	13.7 (0.9)	13.6 (0.7)		13.1 (0.7) ^†^	13.8 (0.8)
Height (cm)	154 (10)	153 (10)	155 (10)		154 (11)	155 (10)
Weight (kg)	46.8 (11.8)	44.5 (11.3)	47.5 (11.9)		42.4 (9.1) ^†^	48.3 (12.3)
BMI (kg/m²)	19.3 (3.3)	18.8 (2.7)	19.5 (3.4)		17.8 (2.3) ^†^	19.8 (3.4)
BMI (z-score)	0.2 (1.5)	−0.2 (1.3)	0.2 (1.6)		−0.3 (1.1) ^†^	0.4 (1.6)
20 m SRT (speed)	9.7 (0.6)	10.1 (0.6) *	9.5 (0.62)		10.3 (0.4) ^†^	9.5 (0.6)
20 m SRT (z-score)	−1.2 (0.5)	−0.9 (0.4) *	−1.3 (0.5)		−0.6 (0.2) ^†^	-1.4 (0.4)
VO2 peak (mL/kg/min)	38.5 (0.5)	40.5 (2.6) *	37.8 (3.5)		42.5 (1.4) ^†^	37.1 (2.9)
Daily MVPA (min)	50.6 (25.5)	75.9 (22.4) *	42.5 (21.3)		56.8 (21.6)	48.4 (21.0)
PE-MVPA (min)	9.9 (3.6)	10.8 (4.0)	9.7 (3.4)		10.3 (3.9)	9.8 (3.4)
PE-MVPA (%)	22.1 (7.9)	23.9 (9.1)	21.5 (7.5)		22.9 (8.6)	21.8 (7.7)

Note. * The mean is significantly different from that of the inactive group (*p* < 0.05). ^†^ The mean is significantly different from that of the unfit group (*p* < 0.05).

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
