# Peer review of "Contribution of Physical Education to the Daily Physical Activity of Schoolchildren in Saudi Arabia"

_ijerph, 2019, doi:10.3390/ijerph16132397_

Round 1
Reviewer 1 Report
Overall, the study is contributing meaningful knowledge in the physical activity levels of students in Saudi Arabia. The following are a few suggestions for the authors to consider:
Introduction, lines 38-44: Perhaps provide more clarity and literature to support the need for the present study.
Abstract: Rephrase this sentence for clarity: “The time that students spent on MVPA accounted for 22 % of the duration of a PE class.” Do you mean 22% of the PE class time is in MVPA?
Abstract: Rephrase this sentence for clarity: “Further, an additional 12.9 minutes (inactive students: 14.7 minutes; unfit students: 14.8 minutes) of MVPA were recorded on PE days than on days without PE.” Is 12.9 minutes the average time in MVPA for all the participants?
Methods, lines 72-73: Why were only weekday data analyzed in this study? Why were the weekend data not used?
Results section, lines 134-135: It may be helpful to provide the percentages of unfit students and inactive students in relation to whole sample.
Figures 1 and 2: What do the percentages above the figures mean? (21%, 16%, 23%). Also, what does the y-axis represent? Please label accordingly.
Discussion, line 196+: Explain why the percentage of engagement on MVPA is lower (22%) in this study. Perhaps providing the context of the PE class would be helpful. What were the students engage in during PE at the time of the study?
Discussion: The authors suggested increasing the number of days of PE in school as an intervention to increase students’ PA? How feasible is this suggestion in the context of the location of the study? An important finding is that students were inactive on non-PE days, therefore, what other suggestions can be incorporated to increase students’ PA levels during non-PE days? Could other forms of non-PE intervention be used (e.g., classroom activities/recess activities, etc?). Also mentioned in lines 282-283.
Discussion, line 266: Explain why boys were the only sample in this study? Why were girls not included in the study?
Author Response
Reviewer 1: Report and response
Introduction, lines 38-44: Perhaps provide more clarity and literature to support the need for the present study.
Response: More clarification and literature were added. Thank you
Abstract: Rephrase this sentence for clarity: “The time that students spent on MVPA accounted for 22 % of the duration of a PE class.” Do you mean 22% of the PE class time is in MVPA?
Response: Rephrasing is done. Thank you
Abstract: Rephrase this sentence for clarity: “Further, an additional 12.9 minutes (inactive students: 14.7 minutes; unfit students: 14.8 minutes) of MVPA were recorded on PE days than on days without PE.” Is 12.9 minutes the average time in MVPA for all the participants?
Response: Rephrasing is done. Thank you
Methods, lines 72-73: Why were only weekday data analyzed in this study? Why were the weekend data not used?
Response: Thank you for your comment. Because of the aim of this study was to examine the contribution of PE to daily PA and to compare weekdays with and without PE, we analyzed weekdays only.
Results section, lines 134-135: It may be helpful to provide the percentages of unfit students and inactive students in relation to whole sample.
Response: Rephrasing is done. Thank you
Figures 1 and 2: What do the percentages above the figures mean? (21%, 16%, 23%). Also, what does the y-axis represent? Please label accordingly.
Response: These values represent the contribution of PE to daily MVPA. They were also reported in the text, so we removed them from figures to avoid confusion. The y-axis represents the time of MVPA in minutes and now is labeled. Thank you
Discussion, line 196+: Explain why the percentage of engagement on MVPA is lower (22%) in this study. Perhaps providing the context of the PE class would be helpful. What were the students engage in during PE at the time of the study?
Response: The possible reasons of low level of MVPA reported in PE lessons are already explained.We did not examine what was the content of PE lesson. Thank you
Discussion: The authors suggested increasing the number of days of PE in school as an intervention to increase students’ PA? How feasible is this suggestion in the context of the location of the study? An important finding is that students were inactive on non-PE days, therefore, what other suggestions can be incorporated to increase students’ PA levels during non-PE days? Could other forms of non-PE intervention be used (e.g., classroom activities/recess activities, etc?). Also mentioned in lines 282-283.
Response: Thank you for your comments. We added other suggestions for increasing PA levels during school time. It could be feasible because of Saudi secondary schools offer five PE classes per week for grade 9 only. Therefore, it can be generalized for all grades. However, Saudi schools do not offer physical activity opportunity during traditional classrooms. They also do not offer recess time. The only opportunity which can be targeted to engage the students in physical activities is the lunch break. However, the limited time (15 min) represents an obstacle as the children usually having their snacks in it.
Discussion, line 266: Explain why boys were the only sample in this study? Why were girls not included in the study?
Response: At the time of the data collection PE classes are offered for boys’ schools only. Thank you
Reviewer 2 Report
In my opinion the word students is wrongly chosen. Maybe the word schoolchildren should be used in the whole paper.
26-31: Add some information about quality of life and about chronic diseases and disorders such as diabetes type 2, obesity, … Also something about “learned young is done old” should be added in this paragraph.
26-31: PA can provide additional benefits for school children in terms of concentration, quality of tasks, ... Can you provide literature regarding this subject?
39: “relatively less fit”: which parameter was used to determine this and what was the cut-off value in this study?
The introduction is quite limited. Something about the different measurement methods of PA and their strengths and limitation should be added.
56: How many girls? How many boys?
82: Which cut-off points were used? Please explain.
85: Can you explain the 20 m shuttle run test? Is this a valid test in schoolchildren?
85: How did you measure the CRF of the excluded students for the shuttle run test?
96: Which cut-off points were used? Please explain.
113: Was normality of data checked?
145: The table should be named + “activity status” and “fitness status” should be placed more to the right
175: Only the results of the total sample are represented here. What about the results of the fit versus unfit on PE days?
212: 50%? In an early paragraph is mentioned that the duration is 45 minutes.
232: Also other benefits such as concentration, better school results, … Is there literature regarding this topic? Please discuss.
252: Please exlain which cut-off point were used in the present study and in the other studies.
260: Is there literature about the variation of intensity in PE lessons? Was there a possibility to control for the intensity of the PE lesson(s) in this study? For example that they got the same or a similar lesson, so that there is no influence of the lesson?
Discussion: Maybe the use of BMI as a health parameter can be discussed. Can you add some literature about this topic and discuss this?
Author Response
Reviewer 2: Report and response
In my opinion the word students is wrongly chosen. Maybe the word schoolchildren should be used in the whole paper.
Response: Thank you. Schoolchildren term was used in the whole paper.
26-31: Add some information about quality of life and about chronic diseases and disorders such as diabetes type 2, obesity, … Also something about “learned young is done old” should be added in this paragraph.
Response: Additional information was added in the text as suggested. Thank you
26-31: PA can provide additional benefits for school children in terms of concentration, quality of tasks, ... Can you provide literature regarding this subject?
Response: Additional information was added in the text as suggested. Thank you
39: “relatively less fit”: which parameter was used to determine this and what was the cut-off value in this study?
Response: Thank you for your comment. In their study, Calahorro-Canada et al (2017) assessed Peak oxygen consumption (VO2peak) using a portable breath by breath metabolic unit while children performing Chester Step Test on a bench. Children and adolescents were classified as fit or unfit according to sex-specific and age-specific references. Plowman and Meredith (2013) Adegboye et al.’s (2011)
The introduction is quite limited. Something about the different measurement methods of PA and their strengths and limitation should be added.
Response: Additional information was added in the text as suggested. Thank you
56: How many girls? How many boys?
Response: All participants were boys. Clarification was added in the text. Thank you
82: Which cut-off points were used? Please explain.
Response: Information about cut-points used are added in the text. Thank you
85: Can you explain the 20 m shuttle run test? Is this a valid test in schoolchildren?
Response: The Procedure of 20 m shuttle run test is explained in the text. Thank you
85: How did you measure the CRF of the excluded students for the shuttle run test?
Response: Thank you for comment. It is mentioned that CRF for students with medical condition was not conducted and they were excluding from the analysis.
96: Which cut-off points were used? Please explain.
Response: More information was added in the text. Thank you
113: Was normality of data checked?
Yes, the normality of data distribution was checked visually for standardized residual using histograms and Q-Q plots and they showed acceptable distribution. Thank you
145: The table should be named + “activity status” and “fitness status” should be placed more to the right
Response: The table already named and was modified as suggested. Thank you
175: Only the results of the total sample are represented here. What about the results of the fit versus unfit on PE days?
Response: Thank you for your comment. The results of the contribution of PE to daily MVPA in fit vs unfit students are already mentioned in the text (daily MVPA = the average PE day + non-PE day)
212: 50%? In an early paragraph is mentioned that the duration is 45 minutes.
Response: Thank you for your comment. 50% means half of the class time which is 45/2.
232: Also other benefits such as concentration, better school results, … Is there literature regarding this topic? Please discuss.
Response: More information was added in the text. Thank you
252: Please exlain which cut-off point were used in the present study and in the other studies.
Response: Thank you. An explanation was added in the text.
260: Is there literature about the variation of intensity in PE lessons? Was there a possibility to control for the intensity of the PE lesson(s) in this study? For example that they got the same or a similar lesson, so that there is no influence of the lesson?
Response: Thank you for your comment. Additional information about the influence of PE content on PA levels is added. The data were collected for two months, and each week a new activity and skills are provided, so it was difficult to control the lessons content as they follow PE plan among all schools. So, we avoided to alter any PE plan as the teacher and students do not know what the main aim of the study was. The only know that data about PA levels will be collected.
Discussion: Maybe the use of BMI as a health parameter can be discussed. Can you add some literature about this topic and discuss this?
Response: Thank you for your suggestion. The limitation of using BMI as health parameter is beyond the scope of the current study. We used BMI as ac covariate variable only.
Reviewer 3 Report
Line 56: I think it is a mistake, mention 123 subjects and in all the paper you will refer to 111, I recommend correction or clarification.
I recommend you mention more clearly when the tests were done. What was the way of selecting the test days and the time when they were done in total.
I recommend that in the Partcicipants section mention the number of subjects in the ages.
Lines 77, repeat idea: for a total duration of seven days, I recommend deleting,
Lines 85: Mention as: Students with medical conditions were excluded from the CRF test, which is their number, at which reference value did you refer to 123 or 111. I recommend clarification.
Lines 96-97. Is not it clear in the end how many subjects have participated in this test? How many were excluded?
Lines 99-101, repeat the idea, here mention at least three days in the previous phrases you refer, repeating to 7 days, I recommend clarification.
Lines 102-103 moved to the participants section.
Lines 177-179 repeat the idea, is enough graphics representation.
Lines 191-192 The idea is repeated in the introduction.
The bibliographic index 34 refers to another age category, making the results presented unreliable.
The limitations of the study are numerous, resulting in very low relevance.
Author Response
Reviewer 3: Report and response
line 56: I think it is a mistake, mention 123 subjects and in all the paper you will refer to 111, I recommend correction or clarification.
Response: Clarification was added. Thank you
I recommend you mention more clearly when the tests were done. What was the way of selecting the test days and the time when they were done in total.
Response: Thank you for your comment. Information about the time of data collection was added in the text in participants section. Regarding the days of test, it was more explained in the procedure section. The duration of data collection was two months and was added in participants section.
I recommend that in the Participants section mention the number of subjects in the ages.
Response: Number of students in each age group were mention in the text as recommended. Thank you
Lines 77, repeat idea: for a total duration of seven days, I recommend deleting,
Response: Repeated words were deleted. Thank you
Lines 85: Mention as: Students with medical conditions were excluded from the CRF test, which is their number, at which reference value did you refer to 123 or 111. I recommend clarification.
Response: Thank you for your comment. Number of excluded students because of medical conditions was added in the text. The initial number of agreed stents to take part in the study were 123 and the final sample was 111 students and this was clarified in the Procedure section from line 102 to line 104 in the original copy.
Lines 96-97. Is not it clear in the end how many subjects have participated in this test? How many were excluded?
Response: Thank you for your comment. It was clarified in the Procedure section from line 102 to line 104 in the original copy.
Lines 99-101, repeat the idea, here mention at least three days in the previous phrases you refer, repeating to 7 days, I recommend clarification.
Response: Thank you for your comment. The repeated idea was deleted.
Lines 102-103 moved to the participants section.
Response: Thank you for your comment. we mentioned the number of final Participants here, because it is related to the inclusion criteria of wearing accelerometer and it is important be mentioned in the data treatment paragraph.
Lines 177-179 repeat the idea, is enough graphics representation.
Response: Thank you for comment. they were deleted from figures to avoid confusion.
Lines 191-192 The idea is repeated in the introduction.
Response: Thank you for your comment. repeated idea was deleted
The bibliographic index 34 refers to another age category, making the results presented unreliable.
Response: Thank you for your comment. only data of students aged 13 years were compared to the results of the current study.
The limitations of the study are numerous, resulting in very low relevance.
Response: Thank you for your comment. Every study has limitations and we are honestly pointing out the study limitations to show that we have considered how they may impact the study finding. However, this is the first study that used accelerometer to monitor PA in students across 7 days and over two months in and out schools.
Round 2
Reviewer 3 Report
no comments